Revisiting Drymaeus germaini (Ancey, 1892) (Gastropoda, Bulimulidae): ecological niche and first anatomical description of a poorly known land snail species from Brazil

Macedo Maria Isabel Pinto Ferreira 1 2
Ovando Ximena Maria Constanza 2
http://orcid.org/0000-0001-6494-309X D’ávila Sthefane 1 2 sthefanedavila@hotmail.com
1 Universidade Federal de Juiz de Fora, Museu de Malacologia Prof. Maury Pinto de Oliveira , Juiz de Fora, Minas Gerais , Brazil
2 Institute of Biological Sciences, Universidade Federal de Juiz de Fora , Juiz de Fora, Minas Gerais , Brazil
Żyła Dagmara
Electronic publication date: 2025 Jul 14
Publication date: 2025
Volume: 13
Electronic Location ID: e19641
Received 2024 Aug 7; Accepted 2025 Jun 2
Copyright: © 2025 Macedo et al.
Copyright year: 2025
Copyright holder: Macedo et al.
License: This is an open access article distributed under the terms of the Creative Commons Attribution License, which permits unrestricted use, distribution, reproduction and adaptation in any medium and for any purpose provided that it is properly attributed. For attribution, the original author(s), title, publication source (PeerJ) and either DOI or URL of the article must be cited.
License URL: https://creativecommons.org/licenses/by/4.0/

Keywords: Atlantic forest, Cerrado, Amazon, Bulimulidae, Conservation

Funding: Fundação de Amparo à Pesquisa de Minas Gerais (FAPEMIG) APQ-01441-21 This work was funded by Fundação de Amparo à Pesquisa de Minas Gerais (FAPEMIG), APQ-01441-21. The funders had no role in study design, data collection and analysis, decision to publish, or preparation of the manuscript.

==============================
Background

Terrestrial gastropods are one of the most imperiled animal groups in the world. However, information on population size and structure, geographic range and their trends over time, which is necessary to assess species conservation status, is unavailable or insufficient for most land snail taxa, making it difficult to apply the IUCN criteria. Ecological niche modelling (ENM) has been used to predict geographic distribution, allowing better characterization of the distribution ranges of endemic or rare species, offering the necessary information for stating their conservation status and planning for conservation measures.

Methods

We compiled occurrence records of Drymaeus germaini and D. suprapunctatus (herein proposed as a junior synonym of D. germaini) associated with museum collections and human observation. A distribution map including geographic boundaries and Brazilian biomes was made with QGis version 3.10.14. For niche modelling, seven bioclimatic variables were used as predictors. The models were performed using different packages in R environment version 4.2.0.

Results and Discussion

We have redescribed Drymaeus germaini based on the inner anatomy and shell sculpture, also providing the first comparative conchological analysis with congeneric species. We also updated the current distribution of the species within the main Brazilian biomes and estimated its potential geographic distribution using the ENM approach. The ENM results revealed a belt of mid-to-high suitability areas from Northeast to South Brazil in the Atlantic Forest, the most degraded biome in Brazil. In the Northeast these areas extended from the Atlantic Forest to the Caatinga and in the South, from the Atlantic Forest to the Pampas. Additional areas of mid-suitability were found in Cerrado, the second most degraded biome in Brazil. Our results also revealed a continuous area of mid-to-high environmental suitability for D. germaini within the Arc of Amazon deforestation. Our results evidence the scarcity of occurrence records for this species and most of these records do not correspond to protected areas. Also the ecologically suitable areas for D. germaini are in regions disturbed by deforestation, fires, urbanization, habitat loss, mining, and flooding, which may represent the main threats for this species. Nonetheless, the information necessary to apply the IUCN criteria for D. germaini is still incipient and this species should be classified as data deficient.

Introduction

Bulimulidae Tryon, 1867 (Orthalicoidea Martens, 1860) comprise a diverse group of land snails distributed in South and Central America, besides adjacent areas in USA and Caribbean (Breure & Borrero, 2019). The current taxonomy considers that Bulimulidae consists of three subfamilies: Bostrycinae Breure, 2012, Bulimulinae Tryon, 1867, and Peltellinae Gray, 1855 (Bouchet et al., 2017; Salvador et al., 2023) and 16 genera (Salvador et al., 2023). Bulimulidae is recognized as a family with around 700 species, the greater diversity of species being concentrated within the area comprising northern South America and the Brazilian territory (Solem, 1969; Breure, 1979). About 120 bulimulid species allocated in 12 genera are currently known to occur in Brazil (Simone, 2006; Salvador, 2019a; MolluscaBase, 2024).

Among the bulimulid genera, Drymaeus Albers,1850 is recorded in the Southern part of North America, Caribbean islands, Central and South America, including the Andes and all mainland forests (Breure, 1979; Simone, 2006). This genus was previously ascribed to the subfamily Bulimulinae Tryon, 1867 (Breure, 1979; Schileyko, 1999), but molecular evidence provided by Breure & Romero (2012) have justified its placement in the subfamily Peltellinae. The number of species within this genus has changed over time. Until the late 1960’s, Drymaeus included about 552 species (Breure, 1979). Nowadays, 379 species of Drymaeus are recognized as valid (MolluscaBase, 2024), the decrease of the number of accepted species in Drymaeus being mainly caused by elevation of the subgenera Mesembrinus and Antidrymaeus to the genera level (Salvador et al., 2023). Despite the impressive number of Drymaeus species, data on internal anatomy, distribution, and life history of these species are scarce for most of the taxa (Macedo, Ovando & D’ávila, 2023).

The original diagnosis of Drymaeus was established by Albers (1850) and it was based solely on shell morphology. Lately, the importance of the anatomical characters for species delimitation was shown by Pilsbry (1897), encouraging the inclusion of information on anatomical traits in species diagnosis by subsequent authors (Breure, 1979; Schileyko, 1999; Simone & Salvador, 2016; Simone & Amaral, 2018). Pilsbry (1897) retained two subgenera, i.e.,: Drymaeus for species with the outer lip expanded or reflected and Mesembrinus for species with the outer lip unexpanded or simple. Drymaeus is characterized by peristome usually expanded; jaw with 13 to 18 plates, which are four to five times as long as wide; a radula with straight tooth rows, with relatively large, mono- to tricuspid teeth and bi- to tricuspid latero-marginal teeth (Breure, 1979). Mesembrinus is characterized by a simple peristome; mandibula with 20 plates, which are eight times as long as wide; transverse rows of radula V- or W-shaped with relatively small tri- to multicuspid central and latero-marginal teeth. In addition to Drymaeus and Mesembrinus, Germain (1907) proposed the subgenus Antidrymaeus to allocate species with sinistral shells previously ascribed to Drymaeus. Recently, Salvador et al. (2023) have elevated Mesembrinus and Antidrymaeus to genus level.

In Brazil, 40 species of Drymaeus, 13 of Mesembrinus and two of Antidrymaeus were recorded (Simone, 2006; Macedo, Ovando & D’ávila, 2023; Salvador et al., 2023), most of them characterized only on the basis of shell morphology; a better resolution of species numbers depending on the clarification of potential synonyms, besides the definition of the taxonomic status of varieties and subspecies and elucidation of complexes of species morphologically cryptic, by using molecular approaches (Breure & Borrero, 2019).

Drymaeus germaini Ancey, 1892 is a species originally ascribed to the genus Bulimulus Leach, 1814, based on a single dry shell, collected in Mato Grosso state, Brazil. In the original description, no exact collection locality was given. Ancey (1892) compared D. germaini with only one species of Drymaeus, i.e., “It is in somewhat like a small, thin B. felix, but otherwise quite distinct from the New Granada shell”. Pilsbry (1898), based on shell morphology (aperture shape, expanded lip, and protoconch sculpture), reallocated D. germaini within the genus Drymaeus. Morretes (1949) ascribed it to the subgenus Mormus Martens, 1860. Later, the presence of shell aperture with expanded lip justified the inclusion of D. germaini in the subgenus Drymaeus (Breure, 1979). So far, no anatomical characterization of D. germaini has been provided.

The current knowledge on the distribution of D. germaini is based on few occurrence records taken from the original description and former literature (Pilsbry, 1897; Ancey, 1901; Morretes, 1949, Simone, 2006). The distribution of several bulimulid species in Brazil, particularly species of Drymaeus and Mesembrinus, is poorly known (Dutra-Clarke & Souza, 1991; Simone, 2006; Macedo, Ovando & D’ávila, 2023). The lack of knowledge regarding the distribution of native bulimulid species constitutes an impediment to evaluating their conservation status and, consequently, to advocate for protective laws for vulnerable species.

Ecological niche modelling (ENM) has been used to predict species geographic distribution, allowing researchers to collect species whose occurrences have been unnoticed for great periods of time and thus better characterize their distribution ranges, offering the necessary information for stating their conservation status and planning for conservation measures (Sillero et al., 2021). Another important application of ENM is related to the potential effects of climate change on biodiversity conservation (Beltramino et al., 2015), enabling the evaluation of the potential impacts over species distributions. This approach allows the development of conservation strategies by revealing areas of critical concern, which is especially useful for species with restricted distributions, i.e.,: endemic or rare species (Beltramino et al., 2015), as may be the case for D. germaini. Despite the possibility of using modelling tools to assess the potential distribution of species and favorable areas for species recollection, this approach has been rarely used (Macedo, Ovando & D’ávila, 2023) and fieldwork to specifically assess the current distributional range of bulimulid species has not been conducted recently in Brazil. One reason for the scarcity of ENM studies on bulimulids may be the insufficiency of occurrence records for the species in literature. In this context, malacological collections constitute an important source of unpublished distributional information for ecological niche modelling with the aim of estimating suitability areas and potential species distribution (Pearson et al., 2007; Ball-Damerow et al., 2019).

In the present study, the bulimulid species D. germaini was revisited. We have redescribed this species based on the inner anatomy and shell sculpture, also providing the first comparative analysis with congeneric species. We also updated the current distribution of the species within the main Brazilian biomes and estimated its potential geographic distribution using the ENM approach.

Materials and Methods

Study area

Brazil has the fifth largest territory of the world and geographically it consists of five regions: North, Northeast, Southeast, South and Central-West. North Brazil corresponds almost entirely to the Amazon Forest domain, which occupies almost half of the country (IBGE, 2019). The equatorial climate, hot and humid, is predominant, with irregular average annual rainfall ranging from 1,500 to 2,500 mm/year. Northeast Brazil includes the semi-arid domain, which has an irregularly distributed average annual rainfall ranging from 250 mm/year to less than 750 mm/year. The Southeast includes tropical areas, receiving most of the annual rainfall during the summer, with the annual average rainfall varying from 1,500 to 2,000 mm/year (IBGE, 2019). South Brazil comprises a temperate zone with cool and relatively dry winters and warm and relatively humid summers. The annual distribution of rainfall over southern Brazil is relatively uniform, the average annual rainfall varying from 1,250 to 2,000 mm/year (IBGE, 2019). The Central-West Region stretches from the fringes of the Amazon basin in the west to the state of Goiás in the east. At its westerly extreme it has a relatively well-distributed average annual rainfall of up to 2,500 mm/year. Further to the east, rainfall decreases to about 1,000 mm/year (IBGE, 2019). The Southern region corresponds to the Brazilian Pampas. It is characterized by a rainy climate, with the annual average of rainfall ranging from 1,250 to 2,000 mm/year, without a systematic dry period, but marked by the influence of polar fronts and negative temperatures in the winter (IBGE, 2019).

Geographic distribution

Occurrence records of D. germaini were obtained from the biodiversity databases Global Biodiversity Facilities—GBIF (https://www.gbif.org) and Sistema de Informação sobre a Biodiversidade Brasileira (https://www.sibbr.gov.br), besides published papers and records associated to specimens deposited in the malacological collection of the Museu de Malacologia Prof. Maury Pinto de Oliveira. We validated species identification associated with the occurrence records, by requesting images of the specimens to the curators of the malacological collections consulted online and by performing a morphological study of the specimens deposited in the CMMPO. We could not validate species attribution associated to the specimens from MNRJ, as they were lost during the fire of September 2018. However, these specimens were previously identified by Dr Norma Campos Salgado, a Brazilian malacologist who has dedicated her life to the study of bulimulid land snails, so we considered that these records are trustworthy. From the total number of occurrence records, four had only “São Paulo” as the collection site. These records were excluded from the analysis, as it was not possible to discriminate if the locality referred to the state of São Paulo, or the city of São Paulo. We also removed all the duplicated records, and thus the remaining twelve unique sites with trustable geographic information were used during the ecological niche modeling. We used the application Gazetteers (https://www.geo-locate.org) for the georeferentiation of the localities for which geographic coordinates were lacking. When necessary, the coordinates were converted to decimal degree format using Instituto Nacional de Pesquisas espaciais—INPE geographic calculator (http://www.dpi.inpe.br/calcula/). We plotted and overlapped the occurrence records of D. germaini with the layers of biomes and administrative boundaries obtained from Instituto Brasileiro de Geografia e Estatística—IBGE (https://www.ibge.gov.br). The data was analyzed using QGis version 3.10.14- La Coruña (QGIS Development Team, 2021).

Ecological niche modeling

To delineate the abiotic components of the model, we used 19 bioclimatic variables obtained from the WorldClim database (http://www.worldclim.org) at a spatial resolution of 30 arc seconds (~1 km2), besides two topographic variables, elevation (in meters) and soil type, also with the spatial resolution of 30 arc seconds (~1 km2), obtained from IBGE (https://downloads.ibge.gov.br). We clipped the environmental data layers to the mask defined as the Brazilian territory (in shapefile format) and a calibration area considered as the polygon formed by the states with records for D. germaini and D. suprapunctatus, using the “raster” package in R version 4.2.0 (R Core Team, 2021) and the QGIS program (3.22.14 version). Multicollinearity among the variables was analyzed using the Variance Inflation Factor (VIF) with the “usdm” package (Naimi et al., 2014). Highly correlated variables were excluded from the previous step to construction of the niche modeling, with the default cutoff set at greater than or equal to 10 to avoid collinearity in statistical models, resulting in seven bioclimatic variables for the analyses: BIO2 = mean diurnal range (mean of monthly (max temp–min temp)), BIO 3 = isothermality ((BIO2/BIO7) (*100)), BIO 8 = mean temperature of wettest quarter, BIO13 = precipitation of wettest month, BIO 15 = precipitation seasonality (coefficient of variation), BIO 18 = precipitation of warmest quarter, BIO 19 = precipitation of coldest quarter, besides elevation. The ENMs were calibrated and performed with the software package MaxEnt version 3.4.3 (Phillips, Dudík & Schapire, 2021), through the R package “kuenm” (Cobos et al., 2019). The candidate models were created by combining three sets of environmental variables (Set 1: all the variables resulted from VIF; Set 2: only the bioclimatic variables, and Set 3: elevation and bioclimatic variables except BIO3), five values of regularization multiplier setting (0.1–1.0 at intervals of 0.3, and 2 at intervals of 1), and the six feature combinations (linear = l, quadratic = q, product = p, lp. lq, lqp). Candidate model performance was evaluated based on significance (partial ROC, with 100 iterations and 70 percent of data for bootstrapping), omission rates (E = 5%), and model complexity (AICc) values of ≤ 2, according to the proposal of (Cobos et al., 2019). Final models were created using the full set of records with ten replicates by bootstrap with 100 iterations, and logistic output. The “logistic” output returned a continuous map with an estimated probability of presence between 0 (no probability of the species presence) and 1 (high probability of presence), which permits fine distinctions between the suitability of different areas modeled. Jackknife tests were used to identify the variables that most influenced the model predictions. The resulting model was evaluated based on significance, i.e., partial ROC and then, the modes were also projected to the limits of Brazil and the response curve was calculated for the variables of the final model.

Morphological study

The specimens used for the morphological analyses came from the malacological collection of the Museu de Malacologia Prof. Maury Pinto de Oliveira, Universidade Federal de Juiz de Fora–Brazil. For the conchological characterization, only intact adult shells were selected. Sixteen shells were photographed with a Digital camera Nikon D5300 and measured according to the methodology proposed by Miranda (2015). The obtained images were used to construct Tps files with the software TpsUtil version 1.81 (Rohlf, 2021) and the shell measurements were carried out with TpsDig2 version 2.32 (Rohlf, 2018) directly on the image with the “Make Linear Measures” tool on apertural and lateral views of the shells. The following linear measurements were taken: total shell height (tsh), body whorl height (bwh), spire height (sh), major shell diameter (masd), minor shell diameter (misd), apertural height (ah), apertural diameter (ad), parietal space length (psl), penultimate whorl height (pwh), and penultimate whorl diameter (pwd). For the anatomical characterization of the reproductive and pallial systems, three adult snails were dissected immersed in ethanol, under an Olympus® stereoscopy microscope, model SZX7. The anatomical structures were drawn with the aid of a camera lucida. Radulae and shells from two adult specimens were prepared for scanning electron microscopy, following the methodology proposed by Ploeger & Breure (1977). The radulae were separated from the buccal mass, cleaned by immersion in sodium hypochlorite solution and dried. The shells and radulae were mounted on stubs covered with carbon tape, metalized with 50 nm gold, and observed under a scanning electron microscope FEI Quanta 250 at the laboratory of Electron Microscopy, Federal University of Juiz de Fora.

Results

Taxonomic part and morphology

Systematic arrangement

Superfamily Orthalicoidea Martens, 1860

Family Bulimulidae Tryon, 1867

Subfamily Peltellinae Gray, 1855

Drymaeus Albers, 1850

Type species: Helix hygrohylaeus (d’Orbigny, 1835)

Drymaeus germaini Ancey, 1901

Bulimulus germaini Ancey (1892): 91.

Drymaeus germaini—Pilsbry (1897): 206.

Drymaeus (Mormus) germaini—Morretes (1949): 150.

Drymaeus (Drymaeus) germaini—Breure (1979): 109.

Drymaeus germaini—Salgado & Coelho (2003): 162; —Simone (2006): 137; —Salvador (2019b): 86.

Drymaeus linostoma suprapunctatus F. Baker (1914): 638.

Drymaeus suprapunctatus F. Baker (1914): 638.

Drymaeus suprapunctatus—Simone (2006): 144.

Type material: Holotype of Drymaeus germaini Ancey, 1901. Brazil • Mato Grosso; UMMZ 124281. Paratype of Drymaeus linostoma suprapunctatus F. Baker, 1914. Brazil • along Madeira-Mamoré railroad 284 km above Porto Velho [camp 39], Rondonia; IZ CAS 64150.

Type locality: Brazil • Matto Grosso [Mato Grosso].

Material examined: Brazil • 1 shell; Minas Gerais, Ponte Nova, Jul. 1961; CMMPO 1865. • 2 shells; Minas Gerais, Ipaba, Fazenda da Macedônia, 06 Apr. 2011; CMMPO 8697. • 3 shells; Minas Gerais, Ipaba, Fazenda da Macedônia, 10 Nov. 2011; Junqueira, F. O. leg.; CMMPO11258. • 2 shells; Minas Gerais, Ipaba, Fazenda da Macedônia, 27 May 2011; Junqueira, F. O. leg.; CMMPO11260. • 2 shells; Minas Gerais, Ipaba, Fazenda da Macedônia, 18 Nov. 2012; Junqueira, F. O. leg.; CMMPO11261. • 8 shells; Minas Gerais, Ipaba, Fazenda da Macedônia, 05 Jul. 2011; Junqueira, F. O. leg.; CMMPO11262. • 1 specimen preserved in ethanol; Minas Gerais, Ipaba, Fazenda da Macedônia, 18 Nov. 2012; Junqueira, F. O. leg.; CMMPO11263. • 3 shells; Minas Gerais, Ipaba, Fazenda da Macedônia, 27 May 2011; Junqueira, F. O. leg.; CMMPO11264. • 1 shell; Minas Gerais, Ipaba, Fazenda da Macedônia, 12 Jul. 2012; Junqueira, F. O. leg.; CMMPO11265. • 3 shells; Minas Gerais, Ipaba, Fazenda da Macedônia, 11 Apr. 2012; Junqueira, F. O. leg.; CMMPO11266. • 6 shells; Minas Gerais, Ipaba, Fazenda da Macedônia, 17 May 2012; Junqueira, F. O. leg.; CMMPO11267. • Minas Gerais, Ipaba, Fazenda da Macedônia, 17 May 2012; Junqueira, F. O. leg.; CMMPO11268. • 1 specimen preserved in ethanol; Minas Gerais, Ipaba, Fazenda da Macedônia, 05 Jul. 2011; Junqueira, F. O. leg.; CMMPO 11269. • 2 specimens preserved in ethanol; Minas Gerais, Ipaba, Fazenda da Macedônia, 28 Apr. 2011; Junqueira, F. O. leg.; CMMPO 11270. • 3 shells; Minas Gerais, Ipaba, Fazenda da Macedônia, 06 Apr. 2011; Junqueira, F. O. leg.; CMMPO 11379. • 2 specimen preserved in ethanol; Minas Gerais, Ipaba, Fazenda da Macedônia, 03 Jul. 2005; Junqueira, F. O. leg.; CMMPO11394. • 1 shell; Minas Gerais, Ipaba, Fazenda da Macedônia, 28 Dec. 2011; Junqueira, F. O. leg.; CMMPO11405. • 1 shell; Minas Gerais, Ipaba, Fazenda da Macedônia, 11 Apr. 2012; Junqueira, F. O. leg.; CMMPO11406. • 1 shell; Minas Gerais, Ipaba, Fazenda da Macedônia, 17 May 2012; Junqueira, F. O. leg.; CMMPO11407.

Information from collections assessed online. Brazil • 5 shells; Minas Gerais, Ponte Nova; 42W53’47.612”, 20S25’0.041”; 20 Jan. 2009; Junqueira, F.O. and Salgado, N.C. leg.; MNRJ 13475. • 2 shells; Minas Gerais, Viçosa, Recanto das Cigarra; 42W51’50.011”, 20S45’30.917”; 20 Jan. 2009; Junqueira, F.O. and Salgado, N.C. leg.; MNRJ 13476. • 3 shells; Minas Gerais, Ponte Nova, Amparo da Serra; May 1961; Costa, C.J.F. leg.; MNRJ 18942. 2 shells; São Paulo, São Paulo; 06 Jul. 1905; Sowerby & Fulton legs. ANSP 89838. • São Paulo, São Paulo; UF 109243. • São Paulo, São Paulo; UF 161247. • 3 shells; São Paulo; NHMUK 1905.4.14.5-7. • 1 shell; São Paulo; FMNH 31322. • 2 shells; Paraná, Casiro; NHMUK 1920.10.19.9-10. • 1 shell; Fazenda Santa Rita de Cássia, Paraná, São João do Caiuá; Salvador, R. B. leg.; MZSP 108007. • 2 shells; • São Paulo; Sowerby and Fulton leg.; RMNH 266069. • 6 shells; São Paulo, São Paulo; UMMZ 124282.

Information from human observation assessed online. Brazil • Trilha dos Jequitibás, Santa Rita Do Passa Quatro, São Paulo state; iNaturalist 361908980. • Trilha dos Jequitibás, Santa Rita Do Passa Quatro, São Paulo state; iNaturalist 361909058. • Alta Floresta, Cristalino Lodge, Mato Grosso state; iNaturalist 178034773. • Sinop, Mato Grosso state; iNaturalist 177871487. • Alta Floresta, Mato Grosso state; iNaturalist 96383503.Brazil • unnamed road, Dourados, Mato Grosso do Sul state; iNaturalist 264391140. • Alta Floresta, Cristalino Lodge, Mato Grosso state; iNaturalist 60894874. • Novo Mundo, Mato Grosso state; iNaturalist1694623.

Morphology

Shell (Figs. 1 and 2, Table 1)

Figure 1 Shell of Drymaeus germaini (Ancey, 1892).

Material examined: Brasil • 1 shell; Fazenda Macedônia, Ipaba, Minas Gerais; 28 Oct 2011; MIPS Macedo & S. D’ávila leg.; F.O. Junqueira col.; CMMPO 11405. (A) ventral view. (B) Dorsal view. (C) Lateral view. Shell height 29 mm.

Figure 2 Scanning electron microscopy images of the shell, radula, and jaw of Drymaeus germaini (Ancey, 1892).

Material examined: Brasil • 2 shells; Fazenda Macedonia, Ipiaba, Minas Gerais; 27 may 2011; MIPS Macedo & S. D’ávila leg.; F.O. Junqueira col.; CMMPO 11260. (A–C). Protoconch. (D–F). jaw. (G–I). radula.

Table 1 Shell linear measurements perimeter and surface area of shells of Drymaeus germaini (Ancey, 1892).

Values given in millimeters.

Measurements	Min-Max (SD)
Mean	
tsh	29.7–25.5 (±1.57)
27.3	
bwh	21.0–17.6 (±1.11)
19.5	
sh	8.8–6.6 (±0.62)
7.8	
masd	10.7–9.1 (±0.49)
10.1	
misd	10.3–7.5 (±0.73)
8.38	
ah	12.84–9.1 (±0.89)
13.6	
ad	9.63–7.0 (±0.68)
8.5	
psl	8.6–6.5 (±0.68)
7.8	
pwh	4.2–3.2 (±0.23)
3.8	
pwd	7.4–6.4 (±0.3)
7.0	
Note:

Legend: ad, shell apertural diameter; ah, shell apertural height; bwh, body whorl height; masd, major diameter of the shell; misd, minor diameter; psl, parietal space length; pwd, penultimate whorl major diameter; pwh, penultimate whorl height; sh, spire height; tsh, shell height; sp, shell perimeter; ssa, shell surface area; spp, spire perimeter; spsa, spire surface area; bwp, body whorl perimeter; bwsa, body whorl surface area; ap, aperture perimeter; asa, aperture surface area.

Shell perforate, oblong-attenuated, shining, very narrowly rimate. Shell height 25.5 to 29.7 mm (Table 1); whorls 61/4, convex, separated by well-marked sutures, the last whorl attenuated, oblong, making a short, sudden ascent at the aperture (Fig. 1). Spire rather obtuse, ~1/3 of the shell length. Transition between protoconch to teleoconch well-marked. Aperture a little oblique, oblong, slightly lunate, angular above, expanded; interior with the parietal wall and columella purple colored. Outer lip broadly reflected, columella slightly curved, dilated at the top. Sutures well marked, simple, sightly oblique. Body whorl bluish white to pale purple with irregularly flexuous or lightning-zigzagged streaks of tawny-bluish, with small punctuations above the suture. Peristome expanded in adult specimens. Body whorl ~ 2/3 of the shell length, rounded; umbilicus narrow, partially covered by peristome reflection; periumbilical area purplish. Protoconch with 1 ½ whorl, bluish white to lilac in color; well-marked with a grating sculpture with axial riblets and spiral striae (Figs. 2A–2C).

Jaw (Fig. 2)

Jaw simple, beige arched; above 10 to 14 pairs of uniform subvertical overlapping plates sold at the inner face, free at their outer edges, shallow horizontal striae; central plate trapezoidal, larger than lateral plates. The plates gradually decrease in size from the center to the border of the jaw (Figs. 2D–2F).

Radula (Fig. 2)

Radula with approximately 82 teeth per row. Mean number of teeth per half row (except for central tooth) 41. Rachidian tooth triangular, slightly smaller than lateral teeth (~9/10 of the length of laterals), symmetric, without cusps (Figs. 2G, 2H); basal plate trapezoidal, tapered, with cutting edge. First lateral teeth triangular, oblique, less tapered than central teeth (Figs. 2G, 2H). A basal-lateral cusp gradually appears in lateral teeth and becomes more distinct in marginal teeth (Fig. 2I). Marginal teeth tricuspid; mesocone well developed with a bicuspid cutting edge; exocone relatively reduced (Fig. 2I). Each radular row is disposed in a shallow V-inverted shape, with rounded borders.

Pallial Cavity

Mantle border simple, thick. Rectum narrow, walls thick. Primary ureter along the rectal side of the kidney up to the top of the pallial cavity. Secondary ureter, long and slender, running along the entire left side of the rectum. Mantle cavity short, ~1/4 of the body whorl. Kidney elongate, narrowly triangular, occupying ~1/3 of pallial cavity length. Pericardial cavity on the left side of the kidney, at the columellar margin of the posterior end of the pallial cavity, with approximately the same length as the kidney. Ventricle triangular. Auricle elongate, ~1/3 of the triangular ventricle. Pulmonary vein converging to pneumostome region, oblique in anterior half of the pallial cavity, running along right edge in the posterior half, and presenting a bifurcation in the last 1/7 of the pallial cavity. Vessels well marked in the right edge of the pallial cavity, in the space between the pulmonary vein and the ureter. Vessels imperceptible in the left edge of the pallial cavity.

Reproductive system (Fig. 3, Table 2)

Figure 3 Anatomy of the reproductive system of Drymaeus germaini (Ancey, 1892).

(A) Dorsal view; (B and C) ventral view. Note: ag, albumen gland; atg, genital atrium; bcd, bursa copulatrix duct; ep, epiphallus; fl, flagellum; ot, ovariotestis; ph, phallus; pr, prostate; sc, spermatheca complex; spo, spermoviduct; vd, vas deferens.

Table 2 Linear measurements of reproductive structures of Drymaeus germaini (Ancey, 1892).

Measurements	Mean (Min-Max)	
Spermoviduct length	25.2 (22.1–27.2)	
Vagina length	2.6 (2.2–3.2)	
Bursa copulatrix duct length	22.2 (20.2–24.2)	
Phallus length	2,5 (2.2–3)	
Epiphalus length	12 (8.4–15.6)	
Flagellum length	3.7 (3.3–4.1)	
Note:

Specimens examined: Brazil • 1 specimen preserved in ethanol; Minas Gerais, Ipaba, Fazenda da Macedônia, 05 Jul. 2011; Junqueira, F. O. leg.; CMMPO 11269 . CMMPO11269. Values given in millimeters. • 1 specimen preserved in ethanol; Minas Gerais, Ipaba, Fazenda da Macedônia, 06 Apr. 2011; Junqueira, F. O. leg.; CMMPO 8692-I. • 1 specimen preserved in ethanol; Minas Gerais, Ipaba, Fazenda da Macedônia, 06 Apr. 2011; Junqueira, F. O. leg.; CMMPO 8692-II. Values given in millimeters.

Albumen gland slightly elongated, connected to the fertilization complex in the distal portion (Figs. 3A, 3B). Free oviduct short and narrow. Spermoviduct long and glandular, measuring 25.2 mm (22.1–27.2) in length (Figs. 3A, 3B). Bursa copulatrix elongated, poorly differentiated from the distal portion of the duct which measures 22.2 mm (20.2–24.2) in length (Fig. 3C). Prostate long, juxtaposed to the uterus, containing numerous tubular acini (Figs. 3B, 3C). Penial complex elongated, with approximately the same length as the spermoviduct when stretched, phallus subcylindrical, measuring 2, 5 mm (2.2–3) in length, penial sheath present (Figs. 3A–3C). Phallus-epiphallus transition undifferentiated externally. Epiphallus measuring 12 mm (8.4–15.6) in length. Insertion of the vas deferens into the epiphallus subterminal. Flagellum present, subcylindrical, short, 3.7 mm (3.3–4.1) in length (Figs. 3B, 3C). Insertion of the bursa copulatrix and penial complex not at the same level. Genital atrium very short, 2.6 mm (2.2–3.2) in length. Spermatheca complex well differentiated, sinuous, lying in superior third of the albumen gland (Figs. 3A, 3B).

Distribution: BRAZIL, Rondônia, Mato Grosso, Mato Grosso do Sul, Goiás, Minas Gerais, Paraná, and São Paulo states. Amazon, Cerrado and Atlantic Forest biomes (Fig. 4).

Figure 4 Occurrence records of Drymaeus germaini (Ancey, 1892) in the Brazilian biomes.

Occurrence data associated with preserved specimens from malacological collections and human observations from iNaturalist database.

Differential diagnose

Shell oblong-attenuated, aperture oblique, slightly lunate. Outer lip broadly reflected. Color bluish white to pale purple with irregularly flexuous or lightning-zigzagged streaks of tawny-bluish. Penis with a sheath, flagellum short, subcylindrical. Genital atrium very short.

Remarks

Considering the congeneric species occurring in Brazil, D. germaini is very similar to Drymaeus suprapunctatus F. Baker, 1914 in shell shape, dimensions and pigmentation pattern. The images provided by Simone (2006) for both species show that the holotype of D. suprapunctatus (ANSP 109307) present, as also observed for D. germaini, a pigmentation pattern characterized by the presence of vertical sinuous stripes and dots. In the image of the D. suprapunctatus ANSP 109307 specimen, a spiral roll of dots is observed from the penultimate whorl up to the shell apex, the stripes being absent. In the image of D. germaini ANSP 89838 specimen, the dots are less evident as they merge with the vertical stripes. The shells of both species imaged by Simone (2006) seem to be identical, except for this subtle difference in pigmentation pattern. Herein, shells of D. germaini collected at the same locality in Minas Gerais evidence the presence of intraspecific variation in shell pigmentation, with a graduation in pigmentation pattern from shells presenting mainly stripes to shells bearing dots predominantly (Fig. 5). The shells also may be more elongate or more rounded. This find indicates that there is no clear hiatus between the patterns presented by the holotype of D. suprapunctatus and the specimen of D. germaini imaged by Simone (2006). We also found images of live specimens in GBIF/iNaturalist, corresponding to D. germaini (Figs. 6A and 6B) showing such variation in pigmentation pattern. From these observations, we are inclined to consider that D. germaini and D. suprapunctatus correspond to a single species and thus we propose D. suprapunctatus to be a junior synonym of D. germaini. This matter must be addressed in a more comprehensive study including the molecular analysis of a geographically representative set of specimens. Nonetheless, we believe that it is important to propose this synonym based on the available diagnostic criteria, to mitigate taxonomic inflation within this genus.

Figure 5 Intraspecific variation in pigmentation pattern of shells of Drymaeus germaini (Ancey, 1892).

(A–E) Penultimate whorl (PW) showing only dots (A), dots and incomplete stripes (B and C), dots and stripes (D) and dots merged with stripes (E). (F–I) Body whorl (BW) showing dots and stripes (F), dots and dots merged with stripes (G), and dots merged with stripes (H and I). One specimen may present different combinations of dots, stripes, and dots merged with stripes throughout the shell.

Figure 6 Living specimens and shells of Drymaeus germaini (Ancey, 1892).

(A–I) Adapted images of living specimens of D. germaini (Ancey, 1892) available online at the Global Biodiversity Database Facility. Shell measurements not provided. (A) Adapted image of a specimen ascribed to D. germaini observed at Trilha dos Jequitibás, Santa Rita Do Passa Quatro, São Paulo state, Brazil. Original image and licence information at https://www.inaturalist.org/photos/361909013 (B) Adapted image of a specimen ascribed to D. germaini observed at Trilha dos Jequitibás, Santa Rita Do Passa Quatro, São Paulo state, Brazil. Original image and licence information at https://www.inaturalist.org/photos/361909118 (C) Adapted image of a specimen ascribed to D. germaini observed at Alta Floresta, Cristalino Lodge, Mato Grosso state, Brazil. Original image and licence information at https://www.inaturalist.org/photos/178034773 (D) Adapted image of a specimen ascribed to Drymaeus suprapunctatus F. Baker, 1914 and herein corrected to D. germaini, observed at Sinop, Mato Grosso state, Brazil. Original image and licence information at https://www.inaturalist.org/photos/177871487 (E) Adapted image of a specimen ascribed to D. suprapunctatus and herein corrected to D. germaini, observed at Alta Floresta, Mato Grosso state, Brazil. Original image and licence information at https://www.inaturalist.org/photos/96383503 (F) Adapted image of a specimen ascribed to D. suprapunctatus and herein corrected to D. germaini, observed at unnamed road, Dourados, Mato Grosso do Sul state, Brazil. Original image and licence information at https://www.inaturalist.org/photos/264391090 (G) Adapted image of a specimen ascribed to D. germaini observed at Alta Floresta, Cristalino Lodge, Mato Grosso state, Brazil. Original image and licence information at https://www.inaturalist.org/photos/60894874 (H) and (I) Adapted image of a specimen ascribed to D. germaini observed at Novo Mundo, Mato Grosso state, Brazil. Original image and licence information at https://www.inaturalist.org/photos/1694623 (J–U) Shells from D. germaini collected at Fazenda Macedônia, Ipaba, Minas Gerais state, Brazil. photographies taken by Sthefane D’ávila. (J–L) CMMPO 11259 specimen 1. Shell height 25 mm. (M–O) CMMPO 11259 specimen 2. Shell height 28 mm. (P–R) CMMPO 11266 specimen 1. Shell height 29 mm. (S–U) CMMPO 11259 specimen 3. Shell height 30 mm. (V-W) NHMUK ZOO 1975206, LECTOTYPE & PARALECTOTYPES, Bulimus felix Pfeiffer, 1862, collected in Venezuela (Bolivarian Republic of) by The Trustees of the Natural History Museum, London, Original images and licence information at https://www.gbif.org/occurrence/1056778445. Shell height ~64 mm. (Y–AA) NHMUK ZOO 1854.12.4.132, SYNTYPES, Helix linostoma d’Orbigny, 1835, collected in Bolivia (Plurinational State of) by The Trustees of the Natural History Museum, London. Original images and licence information at https://www.gbif.org/occurrence/1057483765. Shell height ~59 mm.

The pigmentation pattern of the shell of D. germaini is also similar to the shell of Drymaeus subsimilaris Pilsbry, 1898. Nonetheless, the shells of both species can be distinguished by the characteristics of the sutures and growth lines, D. germaini presenting deeper sutures compared to D. subsimilaris, making the shell whorls more rounded and presenting well marked growth lines, with a distinct horizontal stripe lying under the suture of the body whorl. Additionally, the columellar margin of D. germaini is straighter when compared to D. subsimilaris.

At the time of its description, D. germaini was compared with just one species, D. felix, which was never recorded in Brazil. The shell of D. germaini, as characterized by Ancey (1892), is smaller and thinner than that of D. felix. The observation of the lectotype and paralectotypes of D. felix confirms that the latter species is larger with a very distinct pigmentation pattern, with wide bands, reddish brown, and apex blackish (Fig. 6G). Drymaeus linostoma (d’Orbigny, 1835) is similar to D. germaini, but it presents a bigger shell with a more oval aperture (Fig. 6H). Anatomical data of both species is unavailable. In addition, the distribution of these species does not seem to overlap, at least considering the present knowledge of their geographical ranges.

Ecological niche modelling

Calibration, final models and evaluation

From calibration models, resulting in a total of 108 candidate models, one was statistically significant using the set 2 (only bioclimatic variables) and met with the pre-defined criteria. The final model presented a good performance, with mean AUC values of 0.74, with a standard deviation of 0.06. The Jacknife test of variable importance indicated that the distribution of D. germaini was mainly influenced by the mean diurnal range (BIO 2) (33.7% of contribution) followed by precipitation of coldest quarter (BIO 19) (18.3%), and isothermality (BIO 3) (17.3%) (Table 3). The variable that less contributed to the model was precipitation of wettest month (BIO 13) (2.2%).

Table 3 Relative contribution and permutation importance of environmental variables to the Ecological niche model generated for Drymaeus germaini (Ancey, 1892), in Brazil.

Variable	Percent contribution	Permutation importance	
BIO 2	33.7%	14.9	
BIO 19	18.3%	1.5	
BIO 3	17.3%	23.7	
BIO 18	14.7%	28.9	
BIO 15	7%	26.7	
BIO 8	6.4%	2	
BIO 13	2.5%	2.3	
Note:

WorldClim variables. BIO 2, Mean Diurnal Range; BIO 3, Isothermality; BIO 8, Mean Temperature of Wettest Quarter; BIO 13, Precipitation of Wettest Month; BIO 15, Precipitation Seasonality; BIO 18, Precipitation of Warmest Quarter; BIO 19, Precipitation of Coldest Quarter.

Response curves also gave an indication of the range under which the variable reaches its optimum suitability. The responses of precipitation seasonality (BIO 15), and precipitation of coldest quarter (BIO 19) are ascendent curves with highest values to reach the asymptote, while the response curves of the variables precipitation of warmest quarter (BIO 18), and isothermality (BIO 3) are descendent curves near to zero. Conversely, the variable precipitation of wettest month (BIO 13) and mean temperature of wettest quarter (BIO 8) maintained almost asymptotic behavior (Fig. 7). The optimum suitability of the variable BIO 15 is around 13.3–10.4 mm. Considering the other variables that significantly contributed to the model, the maximum suitability corresponded to isothermality values 45.6–77.4%; mean mean diurnal range temperature (BIO 2) above 7 °C (Fig. 7). Our final data set was composed of 21 localities (Table S1).

Figure 7 Ecological niche modeling for Drymaeus germaini (Ancey, 1892), in Brazil.

(A–F) Contribution of the variables to the model. (G) Environmental suitability for Drymaeus germaini (Ancey), 1892, in Brazil. The ecological niche models were calibrated and performed with the software package MaxEnt version 3.4.3, through the R package “kuenm”.

Estimations of potential geographic distribution

The consensus map with the projection of environmental suitability areas resulting from the ecological niche model obtained herein is shown in Fig. 5. The results of the ecological niche modelling showed areas of mid-to-high suitability for D. germaini in all five Brazilian biomes. The model recovered a narrow belt from Rio Grande do Norte to Rio Grande do Sul states including mid-to-high suitability areas in the Atlantic Forest. At Rio Grande do Norte, Paraíba, Alagoas, and Sergipe states these areas extended from the Atlantic Forest to the Caatinga. In the Rio Grande do Sul state, a continuous area of mid-to-high suitability included both the Atlantic Forest and the Pampas. In the Amazon biome, scattered patches of mid-to-high suitability areas were recovered on East Roraima, East Amapá, North Pará, North Mato Grosso do Sul, and South Pará. A narrow area of mid suitability in Pantanal biome was recovered between Mato Grosso and Mato Grosso do Sul states. Additional areas of mid suitability were found in Cerrado biome, in the Mid-East Goiás, Mid-East Minas Gerais, and North Maranhão state. The areas with higher suitability were recovered in climate zones characterized by high values of rainfall (Atlantic Tropical Weather and Equatorial Weather) or values of rainfall well distributed throughout the year (Subtropical Weather).

Discussion

In the present study, using records from malacological collections, we have updated the geographic distribution of D. germaini, providing, for the first time, specific localities where this species occurs, which also allowed us to estimate its potential distribution. Drymaeus germaini was described from a single dry shell and since its description, occurrence records of this species in the literature are outdated, scarce, and refer only to Brazilian states, with no mention to specific localities. The analysis of the data associated to museum specimens, revealed that D. germaini was collected and then incorporated to a collection for the last time in January 2011. In a recent effort to recollect this species in this same locality, i.e.,: Ipaba, Minas Gerais state, no specimen was recovered (MIPF Macedo, 2023, personal communication). Accordingly, there have been no recent records of this species in the literature. Nonetheless, we found records of D. germaini, and confirmed the species taxonomic identity, in iNaturalist for the following years: 1982; 2015, 2016, 2020, 2022, 2023, and 2024, evidencing that this gap in the entrance of specimens in malacological collections is mainly due to the shortage in collection efforts. We also attempted to find museum occurrence records of D. germaini in Mato Grosso state, where the type locality is enclosed, but we only found records of human observation for localities in Mato Grosso (Alta Floresta, Sinop, and Novo Mundo) and Mato Grosso do Sul (Dourados) (Global Biodiversity Information Facility (GBIF), 2024). These results indicate the need for collecting D. germaini in these localities, improving its representation in malacological collections, and proving a geographically representative set of specimens for molecular studies on species delimitation, population genetics, molecular taxonomy, and phylogeny. Such studies are greatly necessary to better resolve species limits within this genus and their conservation status.

Among the ten environmental variables used in ENM of D. germaini, the main variables that influenced the model were the precipitation of coldest quarter (BIO 19), isothermality (BIO 3), and mean diurnal range temperature (BIO 2). Temperature in known to have a significant impact on the reproductive strategies adopted by various species or populations of land snails, influencing the reproductive success of individuals (Resende, Cardoso & D’ávila, 2020). The risk of desiccation is also an important factor with influence on their life history traits and survival, as land snails depend on environmental moisture for breathing, reproduction, and locomotion (Arad, 1993; Cook, 2001; Storey, 2002).

The ENM results revealed a narrow belt from Rio Grande do Norte to Rio Grande do Sul states including mid-to-high suitability areas in the Atlantic Forest. At Rio Grande do Norte, Paraíba, Alagoas, and Sergipe states these areas extended from the Atlantic Forest to the Caatinga. In the Rio Grande do Sul state, a continuous area of mid-to-high suitability included both the Atlantic Forest and the Pampas. The Atlantic Forest is considered as a biodiversity hotspot with high levels of endemism in both fauna and flora (Myers et al., 2000). It is also the most degraded biome in Brazil, enclosing most of threatened invertebrate species. Additional areas of mid suitability were found in Cerrado biome, in the Mid-East Goiás, Mid-East Minas Gerais, and North Maranhão state. Cerrado is the second most degraded biome in Brazil. The main threats to biodiversity conservation in these two biomes are the habitat fragmentation, fire, and habitat loss as consequence of urbanization pressure, large scale agriculture, and cattle raising (Zaú, 1998; Santana, Delgado & Schiavetti, 2020). These facts highlight the need to monitor D. germaini and obtaining the information necessary to stablish its conservation status. In a previous study on Mesembrinus interpunctus (Martens, 1887) (Bulimulidae, Peltellinae), we observed that this species is present in all six Brazilian biomes, the results of the niche modeling, however, showed only a thin area of high suitability to this species within the Atlantic Forest Biome, indicating that M. interpunctus may be a generalist species, with a niche breadth that might favor its presence in a range of different ecoregions (Macedo, Ovando & D’ávila, 2023). The results obtained for D. gemaini showed a similar pattern for this species. Herein, even though areas of mid-to-high suitability for D. germaini were predominantly found in within the Atlantic Forest domain, these areas were also found within the other five Brazilian biomes. These results seem to indicate that D. germaini may also be a generalist species, probably occurring in a range of different ecoregions.

Herein, D. germaini is reported for the first time for Minas Gerais state, in Ipaba, Ponte Nova, and Viçosa municipalities. Ipaba and Ponte Nova are located at the Vale do Rio Doce, a geographic region near to two large ferruginous geosystems, i.e.,: Serra da Serpentina-Morro do Pilar and Quadrilátero Ferrífero, this last one concentrating major mining companies. In recent years, this area was affected by two environmental disasters caused by the collapse of two dams, i.e.,: Fundão (2015) from Vale Mining Company and Córrego do Feijão (2019) from Samarco Mining Company. These disasters had profound environmental and social effects, with the loss of many human lives, contamination of water bodies and soil with toxic substances, and devastation of forested areas, with severe impacts on the fauna and flora (Carmo et al., 2017; Rotta et al., 2020). Mining is a relevant threat to species conservation, as it can cause habitat loss and fragmentation, resulting in increased extinction risk, particularly to habitat specialists and species with dispersal limitations such as terrestrial gastropods (Pena et al., 2017; Mallett et al., 2021; Macêdo et al., 2024). Another area of high and mid-to-high suitability for D. germaini in Rio Grande do Sul state was affected by a massive flooding, recently, which had a devastating impact, leaving hundreds of towns underwater. The impact of this climatic event on the fauna is still unknown, nonetheless, this flooding may have a huge impact on land snail fauna. In the Amazon Biome, potential threats to the conservation of D. germaini are mainly related to fires and native forest loss. Our results revealed a continuous area of mid-to-high environmental suitability for D. germaini including Southwest Pará and North Mato Grosso, which is part of the Arc of Amazon deforestation. This area includes the microregion of Altamira, which has undergone severe environmental impacts after the implantation of the Belo Monte Hydroelectric Power Plant, resulting in the submersion of extensive areas of native forest. Our results have also showed areas of mid-to-high environmental suitability for D. germaini in East Roraima, East Amapá, and North Pará. These areas have also undergone extensive deforestation, and simulations have shown that Amazon savannization and climate change are projected to increase heat stress risk which could exceed the human adaptation limit by 2,100 (Oliveira et al., 2021).

We observed that none of the occurrence records recovered from malacological collections overlap with federal, state, or private Conservation Unities. Only the records from the municipality of Ipaba, Minas Gerais, correspond to a Private Natural Reserve. Terrestrial gastropods are one of the most imperiled animal groups in the world (Régnier et al., 2015). In Brazil, 62 land snail species are classified as endangered, this number being probably underestimated. The International Union for Conservation of Nature Red List of Threatened Species relies on several criteria, based on population size and structure, geographic range and their trends over time to assess species conservation status. Nonetheless, the necessary information is unavailable or insufficient for most land snail taxa, making it difficult to apply the IUCN criteria (Régnier et al., 2015). Only two species of Drymaeus, i.e., Drymaeus acervatus (L. Pfeiffer, 1857) and Drymaeus henseli (E. von Martens, 1868), were classified, both considered vulnerable (IUCN, 2024); the poor classification of Drymaeus species with regard to its conservation status being primarily due to the lack of data on their abundance and distribution range. This is the case for D. germaini. Our results evidenced the scarcity of occurrence records for this species, most of them corresponding to unprotected areas. Also, the ecologically suitable areas for D. germaini are in regions disturbed by deforestation, fires, urbanization, habitat loss, mining, and flooding, which may represent the main threats for this species. Nonetheless, the information necessary to apply the IUCN criteria for D. germaini is still incipient and this species should be classified as data deficient.

Our analysis also allowed us to critically explore the data associated with museum specimens. Dry specimens made up the majority (86%) of the material contained in malacological collections. Our finds showed the presence of specimens of D. germaini preserved in ethanol in only one malacological collection, i.e.,: Coleção Malacológica Prof. Maury Pinto de Oliveira, evidencing the significance of small regional collections for our knowledge on Biodiversity. Small Natural History collections contribute with fewer specimens than larger, renowned ones, but they offer distinctive knowledge on local biodiversity, filling critical gaps in taxonomy, geographic distribution, and temporal variation of species occurrence (Monfils et al., 2020).

Herein, we observed the presence of intraspecific variation in shell pigmentation of D. germaini, with a graduation in pigmentation pattern from shells presenting mainly stripes to shells bearing dots predominantly. Such variation evidences the absence of a clear hiatus between the shell traits of D. germaini and D. suprapunctatus, which lead us to propose that D. suprapunctatus is a junior synonym of D. germaini. We didn’t have access to ethanol-preserved specimens of D. suprapunctatus. The images of living specimens used herein are the result of human observations that are not associated with voucher specimens. Also, there is no DNA sequence of D. suprapunctatus in molecular data bases. Such analysis depends on the recollection of specimens of populations from Rondonia, Brazil (type locality of D. suprapunctatus). Thus, molecular tools must be used in the future, from the recollection of specimens of D. suprapunctatus and D. germaini, besides anatomical sources of evidence to delimitate or synonymize these species. Subtle differences in shell characteristics, including the pigmentation pattern, have long been used as operational criteria for the delimitation of Drymaeus species, without considering intraspecific variation in these traits (Macedo, Ovando & D’ávila, 2023). Intraspecific variations in color patterns have already been documented for Drymaeus (Breure & Borrero, 2019), however, most of the nominal species within this genus are known only from their shells, the limit between intra- and interspecific variation in shell morphology and pigmentation being unknown. This evidences the need to incorporate other diagnostic criteria in addition to shell traits for the delimitation of Drymaeus species.

Conclusions

Despites the scarcity of recent occurrence records for D. germaini, the fact that the records available correspond to unprotected areas (with the exception of the records from Ipaba, MG), and the fact that the ecologically suitable areas for D. germaini are located in regions disturbed by deforestation, fires, urbanization, habitat loss, mining, and flooding, this species should be classified as data deficient, according to the criteria of the IUCN.

Supplemental Information

Supplemental Information 1 Occurrence records of Drymaeus germaini (Ancey, 1892) obtained from the biodiversity databases.

The application Gazetteers (https://www.geo-locate.org) was used for the georeferentiation of the localities. When necessary, the coordinates were converted to decimal degree format using Instituto Nacional de Pesquisas espaciais ˗ INPE geographic calculator (http://www.dpi.inpe.br/calcula/).

We acknowledge museum curators and collection managers for providing images of D. germaini. Pedro Loureiro, Laboratório de Microscopia Eletrônica da Universidade Federal de Juiz de Fora, Brazil, for his support during SEM Images acquisition.

Institutions acronyms

ANSP Academy of Natural Science of Philadelphia

CMMPO Coleção Malacológica Maury Pinto de Oliveira

FMNH Field Museum of Natural History

IBGE Instituto Brasileiro de Geografia e Estatística

MNRJ Museu Nacional do Rio de Janeiro

MZSP Museu de Zoologia da Universidade do Estado de São Paulo

NHMUK Natural History Museum

UF Florida Museum of Natural History

RMNH Rijksmuseum voor Natuurlijke Historie

UMMZ University of Michigan Museum of Zoology

Additional Information and Declarations

Competing Interests

The authors declare that they have no competing interests.

Author Contributions

Maria Isabel Pinto Ferreira Macedo conceived and designed the experiments, performed the experiments, analyzed the data, prepared figures and/or tables, and approved the final draft.

Ximena Maria Constanza Ovando performed the experiments, prepared figures and/or tables, and approved the final draft.

Sthefane D’ávila conceived and designed the experiments, performed the experiments, analyzed the data, prepared figures and/or tables, authored or reviewed drafts of the article, and approved the final draft.

Data Availability

The following information was supplied regarding data availability:

Raw data is available in Tables 1–3 and as a Supplemental File.

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
