# Peer review of "Revisiting Drymaeus germaini (Ancey, 1892) (Gastropoda, Bulimulidae): ecological niche and first anatomical description of a poorly known land snail species from Brazil"

_PeerJ, doi:10.7717/peerj.19641_

## Round 0.1 · original submission · Major Revisions

· Academic Editor

Major Revisions

Please, take into account all reviewers' comments. Especially the species delimitation issue should be considered.

Reviewer 1 ·

Basic reporting

The authors present a re-description of Drymaeus germaini with high quality figures and analysed its distribution and conservation status using ecological niche modelling.

The locality list in the text (lines 246-280) is not complete. For example, there are specimens from Matto Grosso (the type region!) figured in Supplementary Fig. 1, which are not listed. Please compile a complete list of localities from which you have seen at least photos of specimens that can be identified as D. germaini (perhaps as a supplementary file) and provide (approximate) geographical coordinates for all localities.

It is not clear why so few localities (and only in Mata Atlantica) are shown in Figure 4. Show all localities of D. germaini, which can be located, on the map and use them for environmental modelling.

Experimental design

A critical issue is the delimitation of D. germaini. The authors state that the shells of D. germaini and D. suprapunctatus "seem to be identical" (line 356). Therefore, they have to synonymize the two taxa and also consider the records of D. suprapunctatus in the environmental modelling and the assessment of the conservation status. Of course, it is meaningful to check the delimitation of the species also using genetic data. However, as long as only shell data are available, the delimitation has to be based on shell data. It makes no sense to exclude specimens that cannot be distinguished from D. germaini only because somebody wrote a different name on a label. If the authors think that the delimitation of the species is not possible with shell data, it makes no sense to model the distribution based on a subset of the records identified as D. germaini based on shell data. If they think that D. germaini and D. suprapunctatus cannot be distinguished, they should synonymize the names (until a genetic analyses shows that more than one species is involved).
The authors should state their opinion about specimens identified as D. germaini (and D. suprapunctatus) in iNaturalist also in iNaturalist, at least if they disagree with the current identifications. This would makes their taxonomic opinions more transparent and results in corrections of misidentifications, also in GBIF.
State also how D. germaini can be distinguished from D. subsimilaris Pilsbry, 1898.

Validity of the findings

The unclear delimitation of D. germaini affects the validity of all findings. A clear discussion of the delimitation of the species is necessary.

Additional comments

Line 64. Note that the decrease of the number of accepted species in Drymaeus was mainly caused by the splitting the genus and accepting Mesembrinus and Antidrymaeus as distinct genera.
Line 70. Pilsbry (1897) hardly used anatomical characters for species delimitation (because of the lack of such data for most species).
Line 89, 237, caption of Table 1, 2, Figure 1-6. Add parentheses around "Ancey, 1892".
Line 91, 96, 97, 153. Replace "germaine" by "germaini".
Line 132-151. Shorten this paragraph and focus on the range of Drymaeus germaini. This paper is about this species and not about the country Brazil.
Line 245. Data about the type material of D. germaini are missing. State whether you tried to locate the type specimens of the species (where?) and whether you think that the type material is lost.
Line 288. Replace "paultous", which is difficult to understand here.
Line 297-337. Specify in how many specimens jaw, radula, genitalia, etc. were examined.
Line 327-337. Provide a table with measurements of the copulatory organs. Such measurements may be species specific and they represent important raw data. Approximations ("~1/6", "~1/8") are not exact enough.
Line 390. Replace "4000-4500" by "4000-4500 mm".
Replace Figures 1 and 6 by Supplementary Fig. 1 (add specimens from Figs 1 and 6, if there are additional variants) in the main text. It is important to show the variation of the species (also for the discussion of the status of D. suprapunctatus).

·

Basic reporting

The study is a comprehensive work that revises the distribution, taxonomy and future population status of the terrestrial snail Drymaeus germaini. The authors provided really extensive efforts in samplings of museum specimens, collection of data and detailed statistical analysis for ecological modeling. I liked the comprehensive style of the manuscript, but there are two major issues: Firstly, the absence of molecular markers, even in discussion of previous findings, if any, is a remarkable drawback, whereas secondly, after such a wide ecological analysis, I would expect a more precise inference to be proposed regarding the conservation status. I recommend the authors to fix these two issues taking into consideration the following comments as well, and accordingly modify the manuscript.
Lines 27-30. I believe the authors should include some details (1 sentence) for the methodology used concerning anatomical and morphometric analyses. They are directly mentioned in Results but nothing is written in Materials and methods to explain to the reader what has been done
Lines 42-44. This is a very general statement, something more specific would be appreciated. For instance, it should be included in the IUCN red list of threatened species.
In introduction the authors should include more data regarding molecular taxonomy and what was changed after initial morphological inferences.
Some of the study area details transfer to introduction
Line 347 - Remarks
Are there any inferences regarding the phylogenetic relationships of the two species, D. germaini and D. suprapunctatus? Do molecular data distinguish clearly the two species. This info has to added and discussed at this point. The authors mention that morphological data support classification of the two species in one single taxon (lines 364-365) and correctly mention that molecular analysis is suggested to confirm this issue, but aren’t any molecular analyses performed so far?
Line 470: Any suggestion for this difference? Where can this be attributed? Maybe some evolutionary proposals could be added here
Lines 504-514: I believe this paragraph does not fit here. It should be better be transferred in the introduction

Experimental design

The major drawback is the lack of molecular analysis.

Validity of the findings

The taxonomic findings are interesting and seem to be supported, but again have o be supported by molecular markers.
Additionally, I would expect after such a wide ecological study a more precise inference concerning the conservation status of Drymaeus germaini in Brazil

·

Basic reporting

The manuscript is clearly presented. There are some errors in English language usage which could be improved by having a native English speaker review the manuscript. I have made some comments throughout the manuscript, mainly correcting small errors of language usage, but these should not be considered exhaustive.
The introduction is thorough with enough information to give a good context to the project. It is suitably referenced. The structure of the paper conforms to the PeerJ standards. The figures are of high quality with good labelling. All data are included in the manuscript.

Experimental design

This manuscript contains original primary research within the scope of the journal. The research question is well defined and relevant, and fills an existing knowledge gap. The methods are described thoroughly and are replicable, and the investigation is thorough and performed to a high standard.

Validity of the findings

The authors present their data clearly; their data are statistically sound and robust. All conclusions are clearly stated and are supported by the results.

Additional comments

This paper is a useful contribution to the scientific literature in this poorly known group, clarifying the description of the focus species and demonstrating how desktop applications can be used to determine suitable areas to focus collection efforts. I would like to know if the species is also found outside Brazil, and if so, where else it is recorded from - which is relevant to its overall conservation status.

---

## Round 0.2 · Minor Revisions

· Academic Editor

Minor Revisions

Please, correct the issues pointed out by one of the reviewers.

Reviewer 1 ·

Basic reporting

I fully agree with the authors that records referred only to Brazilian states should not be used. However, if you want to model the distribution area of a species for which only few identifiable records (museum specimens or iNaturalist records) are available, all localisable sites should be used and listed with coordinates. You cannot enter the locations on the map if they do not have coordinates. Why do you want to withhold the coordinates from readers and future researchers? Original coordinates may be more exact than the coordinates that you can retrieve from a map with just a locality name. Please list all localisable sites with coordinates. It is one of the basic functions of such a revision to summarise all available data.

It is correct that "traditionally, morphometric data of anatomical structures are not used as operational criteria for delimitation of Drymaeus species". It is correct that you cannot use these data if there are no such data from other species. You provide a good redescription of D. germaini and you have dissected the genitalia. It would be easy for you to measure the genitalia and provide the data for the next one who describes a similar species and will be thankful for your data because it allows him to evaluate and specify morphometric differences.

I guess you will be against using genetic data, because "traditionally, DNA sequences are not used as operational criteria for delimitation of Drymaeus species"?

As a taxonomist you are a data provider for many other branches of biology. Do your job and
- list all locality data of the species you have revised with coordinates
- correct misclassifications on iNaturalist. On iNaturalist everyone can see that you have correctly identified the specimens. No one can steal the "originality of our work" (we are not talking about new species). You can even use iNaturalist to promote your upcoming paper. You owe it to the citizen scientists to give them and the scientific community (the iNaturalist data will end up in GBIF) correct identifications.
- provide the measurements of the genitalia of the specimens you dissected.

This is a minor revision. I would recommend that the editor accepts your paper once you show that you have addressed these points. Please save yourself, the editor and the referees the time that another round will take if you start arguing again that this or that is not necessary.
I look forward to reading the published paper.

Experimental design

-

Validity of the findings

In the conclusions, you now suggest to classify the species as vulnerable without having discussed the criteria the IUCN requires for classifying a species as Vulnerable (it has to meet any of the criteria A to E; see IUCN regulations). This widespread (but under-recorded) species is certainly not vulnerable. It is probably of Least Concern, but might be classified as Data Deficient. Delete this from the conclusion and the abstract or discuss the IUCN criteria thoroughly.

·

Basic reporting

The revised version of the study is substantially improved. The absence of molecular data is a general drawback, which however, at this point can be omitted

Experimental design

Again, apart from the lack of molecular markers, the remaining parts are valid

Validity of the findings

The proposed taxonomic findings are valid

·

Basic reporting

The manuscript has been improved by the revision, with errors in English usage corrected and the concerns of reviewers well addressed.

Experimental design

No comment.

Validity of the findings

I am glad to see the status of D. suprapunctatus resolved in this version.

---

## Round 0.3 · accepted · Accept

· Academic Editor

Accept

I am happy with the current version of the manuscript and recommend it for publication.